# Dynamic Changes in Hepatitis A Immunity in Regions with Different Vaccination Strategies and Different Vaccination Coverage

**DOI:** 10.3390/vaccines10091423

**Published:** 2022-08-29

**Authors:** Karen K. Kyuregyan, Maria A. Lopatukhina, Fedor A. Asadi Mobarkhan, Vera S. Kichatova, Ilya A. Potemkin, Olga V. Isaeva, Anastasia A. Karlsen, Elena Yu. Malinnikova, Alla N. Kaira, Tatyana V. Kozhanova, Victor A. Manuylov, Elena P. Mazunina, Evgeniia N. Bykonia, Denis A. Kleymenov, Margarita E. Ignateva, Olga E. Trotsenko, Anna V. Kuznetsova, Anna A. Saryglar, Natalia D. Oorzhak, Victor V. Romanenko, Mikhail I. Mikhailov

**Affiliations:** 1Department of Viral Hepatitis, Russian Medical Academy of Continuous Professional Education, 125993 Moscow, Russia; 2Laboratory of Viral Hepatitis, Mechnikov Research Institute of Vaccines and Sera, 105064 Moscow, Russia; 3Scientific and Educational Resource Center for High-Performance Methods of Genomic Analysis, Peoples’ Friendship University of Russia (RUDN University), 117198 Moscow, Russia; 4Department of Neurology, Neurosurgery and Medical Genetics, Faculty of Pediatrics, Pirogov Russian National Research Medical University, 117997 Moscow, Russia; 5Chumakov Federal Scientific Center for Research, Development of Immunobiological Products, Russian Academy of Sciences, 108819 Moscow, Russia; 6Translational Biomedicine Laboratory, Gamaleya National Research Center for Epidemiology and Microbiology, 123098 Moscow, Russia; 7The Sakha Republic (Yakutia) Regional Department of Rospotrebnadzor, 677027 Yakutsk, Russia; 8Khabarovsk Research Institute of Epidemiology and Microbiology, 680000 Khabarovsk, Russia; 9Center for the Prevention and Control of AIDS and Infectious Diseases, Health Ministry of Khabarovsk Region, 680031 Khabarovsk, Russia; 10Hospital of Infectious Diseases, 667003 Kyzyl, Russia; 11Tuva Regional Department of Rospotrebnadzor, 667010 Kyzyl, Russia; 12Medical Faculty, Ural State Medical University, 620014 Yekaterinburg, Russia; 13Medical Faculty, Belgorod State National Research University, 308015 Belgorod, Russia

**Keywords:** hepatitis A, HAV, herd immunity, hepatitis A vaccination

## Abstract

The data on hepatitis A virus (HAV) seroprevalence are critical for the implementation of a universal mass vaccination (UMV) strategy. The latter has not been implemented in Russia; however, regional child vaccination programs have been adopted in some parts of the country. The aim of this study is to assess changes in HAV immunity within the last decade in regions of Russia with different vaccination strategies and different vaccination coverage rates. In regions where UMV has not been implemented and HAV vaccination coverage rates do not exceed the national average, the 50% seroprevalence threshold has shifted in the Moscow region from people aged under 40 years in 2008 to people aged over 59 years in 2020, and from people aged under 30 years to people aged over 40 years in the Khabarovsk region. In two regions (Yakutia and Sverdlovsk), a two-dose-based UMV scheme has been in place since 2011 and 2003, respectively, and in Tuva single-dose child immunization was launched in 2012. These regional programs have resulted in a significant increase in HAV seroprevalence in children and adolescents. In Yakutia, 50% herd immunity had been achieved by 2020 in age groups under 20 years, compared to 20–30% seroprevalence rates in 2008. In the Sverdlovsk region, HAV immunity has increased to >65% over the decade in children aged over 10 years, adolescents and young adults, whereas it declined in older age groups. However, a three-fold drop in HAV immunity has occurred in children under 10 years of age, reflecting a significant decline in vaccination coverage. In Tuva, HAV immunity rates in children under 10 years old increased two-fold to exceed 50% by 2020. These data suggest that UMV should be implemented on a national level. Measures to control vaccination coverage and catch-up vaccination campaigns are recommended in order to maintain the effectiveness of existing HAV vaccination programs.

## 1. Introduction

Hepatitis A virus (HAV) is a leading cause of acute viral hepatitis worldwide. It is transmitted mainly by direct contact with an infected person or by ingesting contaminated water or food. HAV infection is frequent in countries with poor socioeconomic conditions, where people are exposed to the virus early in life, resulting in frequent asymptomatic infections and a high proportion of immune adults. Due to limited access to quality drinking water, the HAV prevalence is higher in rural areas in high endemic countries [1]. HAV infection can be prevented by means of vaccination. A universal mass vaccination (UMV) strategy is recommended for areas where HAV endemicity is transitioning from high to intermediate levels [2] and is proven to be effective in terms of achieving herd immunity and the elimination of clinical disease [3].

Over the past two decades, HAV endemicity in Russia has shifted from high-to-intermediate to low, as reflected in the gradual decrease in reported incidence rates from 79.4 per 100,000 in 2001 to 1.3 per 100,000 in 2021 [4,5]. Despite the drop in the number of infections in recent years, the incidence of hepatitis A in Russia still retains the general patterns inherent in previous years: multi-year and seasonal cyclicity, increased incidence among adolescents and adults, and geographical unevenness [6]., Improvements in sanitation and access to clean water were apparently the most significant factors contributing to the reduced circulation of HAV in the country, given that a UMV has yet to be implemented in Russia. Since 2001, HAV vaccination has been included in the Russian national immunization schedule only for professional risk groups (healthcare workers, at-risk laboratory workers, sewage workers, workers in institutions that care for people with mental or behavioral disabilities, workers in childcare institutions, canal workers, food handlers) and travelers, and for the purpose of outbreak control [7]. However, a number of regional HAV vaccination programs have been adopted in Russia within the last decade. In chronological order, these are: Two-dose vaccination in the Sverdlovsk region for children aged 6 years since 2003, expanded in 2008 to include all children over 20 months old [8]; two-dose vaccination in the Moscow region for children aged 3 to 6 years before entering kindergarten since 2009 [9]; two-dose vaccination in Sakha Republic (Yakutia) for children aged 20 months since 2011 [10]; and single-dose vaccination in Tuva Republic for children aged 3 years and over since 2012 [11]. However, no comprehensive analysis of the vaccination coverage in these regional programs has been performed so far.

The decrease in HAV circulation among children, who are often asymptomatic and are the primary reservoir of the infection and source of virus transmission to adults, results—in the absence of a UMV strategy—in waning seropositivity in the population and an increase in the proportion of susceptible adults [12]. In turn, this may be followed by an increase in the number of clinically apparent and severe cases of hepatitis A, as the severity of the disease is greater and fatal outcomes more likely in older age groups [13]. Reported incidence data alone is not enough if we are to understand changes in the epidemiology of hepatitis A or estimate the proportion of the population susceptible to infection, as HAV infection may be asymptomatic, especially in children [14]. To answer these questions and to assess the impact of existing prevention programs, data on the prevalence of anti-HAV antibodies in the population are needed. These antibodies persist for life post infection or at least for several decades after vaccination [15]. However, the data on HAV seroprevalence in Russia are fragmentary and need updating. The aim of this study was to assess changes in HAV immunity within the last decade in regions of Russia with different vaccination strategies and different vaccination coverage rates.

## 2. Materials and Methods

### 2.1. Study Cohorts

This cross-sectional study included the population of five geographically distant regions of the Russian Federation, from west to east: the Moscow region, the Sverdlovsk region, Tuva Republic, Sakha Republic (Yakutia), and the Khabarovsk region. Each of these has different climate, economic, social, and demographic conditions, as obtained from official state statistical reports [16], as well as different HAV vaccination strategies (Figure 1).

Sampling was conducted in 2008 and 2020/2021 among independent cohorts of healthy volunteers. The population sample size was calculated for the known size of the population of the study region based on the data relating to anti-HAV antibody prevalence rates in some parts of Russia before 2008 [4] with the chosen power (80%) and confidence level (95%) [17]. The healthy volunteers were persons undergoing routine medical examinations, visitors to the vaccination center undergoing routine vaccinations, and patients visiting the polyclinic for reasons not related to infectious disease. Inclusion criteria were permanent residence in the study region and a signed and dated informed consent form approved by the Ethics Committee. Exclusion criteria were as follows: children in care, treatment with blood products within 3 months before entering the study (self-reported or parent-reported), and a body temperature over 37.10 °C or acute illness.

The study was conducted in accordance with the principles laid out in the World Medical Association’s Declaration of Helsinki regarding ethical medical research involving human subjects. Written informed consent was obtained from all participants or their parents (or legal guardians). The study design was approved by the Ethics Committee of the Chumakov Institute of Poliomyelitis and Viral Encephalitides, Moscow, Russia (Approval #6 dated 1 April 2007) and the Ethics Committee of the of the Mechnikov Research Institute for Vaccines and Sera in Moscow, Russia (Approval #1 dated 28 February 2018).

The distribution of samples between regions and years of the study is shown in Figure 1. In 2008, a total of 5237 blood serum samples were obtained from healthy volunteers in the five study regions and separated into the following nine age groups: <1 year, 1–9 years, 10–14 years, 15–19 years, 20–29 years, 30–39 years, 40–49 years, 50–59 years, and 60+ years. In 2020/2021, a total of 15,007 sera were collected in the same regions from healthy individuals representing the same nine age groups, except for Moscow and Khabarovsk regions, where no samples from children under 1 year old were collected. In the Moscow and Khabarovsk regions, the sample size was increased compared to the 2008 study in order to retain the power of analysis due to the low HAV seroprevalence rates observed in the 2008 cohorts. The numbers of participants in study cohorts and their sociodemographic characteristics are shown in Table 1.

All serum samples were shipped to the laboratory using cold chain, coded, and stored in aliquots at −70 °C until testing. After thawing, sera were tested for anti-HAV antibodies as described below.

### 2.2. Anti-HAV Antibody Testing

Sera collected in 2008 were tested the same year for anti-HAV IgG antibodies using the Enzyme-Linked Immunosorbent Assay (ELISA) Monolisa Total Anti-HAV Plus Kit (BioRad, France). Samples were considered positive when the concentration of anti-HAV antibodies measured ≥20 mIU/mL. Sera collected in 2020/2021 were tested for anti-HAV IgG antibodies using the commercially available ELISA Vectohep A-IgG kit (Vector-Best, Novosibirsk, Russia) with performance characteristics similar to the assay used for testing in 2008, applying the same positivity criterion of ≥20 mIU/mL. Testing was performed according to the instructions provided by the manufacturers of the respective kits.

### 2.3. Incidence and Vaccination Coverage Analysis

Data on the annual incidence of hepatitis A between 2004 and 2021 in the study regions and in Russia on average were retrieved from the annual federal statistical forms 1, issued by the Russian Federal Service for Surveillance on Consumer Rights Protection and Human Wellbeing (Rospotrebnadzor), on infectious and parasitic disease morbidity.

As data on vaccination coverage (i.e., the proportion of those vaccinated among those who should be vaccinated) in the Russian regions were not available from official reports, the absolute numbers of those vaccinated against hepatitis A per year in the total population and in children under 18 years were retrieved from Rospotrebnadzor’s annual federal statistical forms 5, containing information on vaccinations, and were used to calculate vaccination coverage per 100,000 of the population per year based on the total and child population in each study region according to the Russian Statistics Agency [15]. For the Moscow region, incidence and vaccination coverage rates for Moscow City and the surrounding Moscow region were analyzed together. Although separate federal entities, these are often considered as a single Moscow metropolitan area.

### 2.4. Statistical Analysis

Data analysis was performed using graphpad.com (accessed on 3 June 2022). Statistical analysis included the calculation of a 95% confidence interval (95% CI) for anti-HAV antibody prevalence data and assessment of the significance of differences between groups using Fisher’s exact test (significance threshold *p* < 0.05).

## 3. Results

### 3.1. Hepatitis A Seroprevalence

Average anti-HAV IgG antibody detection rates in the study regions are shown in Table 2. A significant drop in average anti-HAV antibody prevalence occurred in the Moscow and Khabarovsk regions between 2008 and 2020, whereas in Yakutia there was a significant increase in average anti-HAV antibody prevalence over the same period. In Tuva and the Sverdlovsk region, average seroprevalence rates remained stable. Similar trends, though not always statistically significant, were observed in the study regions when data were analyzed separately by gender and place of residence (urban vs. rural).

When analyzed by gender, anti-HAV antibody detection rates in the 2008 cohorts were significantly higher in women compared to men in four out of the five study regions. Only in the Moscow region was the distribution of anti-HAV seropositivity similar between men and women (Table 2). In the 2020/2021 cohorts, significant gender-specific differences in anti-HAV seropositivity rates remained only in Tuva, where women were more often seropositive than men, and in Yakutia, where seropositivity rates were higher in men.

Significant differences in anti-HAV seropositivity rates between urban and rural populations were observed in different years, in four of the regions. In Tuva, the rural population had higher anti-HAV antibody rates in 2008, whereas in 2020 seropositivity rates were higher in the urban population. In the Moscow region, seropositivity rates were similar in 2008, but were higher in the rural population in 2020. In Yakutia and the Khabarovsk region, anti-HAV antibody prevalence rates were higher in the urban population in 2008, but these differences had disappeared by 2020 (Table 2).

Age-specific anti-HAV antibody detection rates are shown for each study region in Figure 2 and in Appendix A. Data for the Moscow region (Figure 2A) are presented together with archived data on age-specific anti-HAV prevalence rates in Moscow in 1981 and 1993. The latter are added for illustrative purposes only, as no direct comparison with data from the current study is possible due to the lack of primary data and information relating to the performance characteristics of the anti-HAV ELISA kits used for testing in 1981 and 1993. Taken together, these data demonstrate a continuous decrease in HAV seroprevalence in the Moscow region from the 1980s until the present day, with a significant decrease in anti-HAV detection rates in all age groups starting from those aged 10–14 years to those over 60 years when the 2008 and 2020 cohorts are compared (Figure 2A). A similar pattern can be seen in the Khabarovsk region, where a significant decrease in HAV seroprevalence was observed in young adults aged 20–39 years (Figure 2E). Consequently, the 50% seroprevalence threshold (i.e., the age at the midpoint of population immunity) in the Moscow region has shifted from those aged under 40 years in 2008 to those over 59 years in 2020, and in the Khabarovsk region from those under 30 years to those over 40 years.

A different pattern was observed in the Sverdlovsk region, Tuva, and Yakutia, where HAV seroprevalence rates in children and/or adolescents had increased by 2020/2021. In the Sverdlovsk region, where child vaccination began in 2003, the peak anti-HAV detection rates shifted from children under 10 years in 2008 to the 10–14 years and 15–19 years age groups in 2021 (Figure 2B). Interestingly, anti-HAV antibody prevalence in this region had significantly declined in children under 10 years by 2021.

In Tuva, HAV seroprevalence significantly increased in children under 10 years, but dropped in adolescents and young adults under 40 years of age (Figure 2C). In Yakutia, anti-HAV antibody detection rates were significantly higher in children, adolescents, and young adults under 30 years in 2020 compared to 2008 but remained the same in older age groups (Figure 2D).

Anti-HAV detection rates in children under 1 year old varied from 22% to 77% depending on the region and year, reflecting changes in HAV seroprevalence among women of childbearing age.

### 3.2. Hepatitis A Incidence and Vaccination Coverage

To assess the impact of changes in hepatitis A morbidity and vaccination coverage on HAV seroprevalence, we analyzed both these parameters in the study regions.

The reported hepatitis A incidence rates in the study regions from 2004 to 2021 in comparison to the Russian national average are shown in Figure 3. Since 2008, there has been a decrease in reported incidence rates to below 10 per 100,000 on average across the country and in all study regions except Tuva. The latter had the highest HAV incidence rates in Russia until 2013, the year following the implementation of the child vaccination program. Since then, incidence in the study regions has continued to decline and has not significantly exceeded the national average, with the exception of two flares: in 2014 in the Sverdlovsk region and in 2017 in the Moscow region.

Calculated HAV vaccination coverage rates in the study regions, expressed as the number of those vaccinated per 100,000 per year, are shown in comparison to the national average in Figure 4. The total numbers of those vaccinated per region per year are shown in Appendix A. Despite the child vaccination program announced in Moscow in 2009, the annual coverage rates in this region do not exceed those in Khabarovsk, where no child vaccination program is in place. In these two regions, annual HAV vaccination rates are similar to the national average and reflect low levels of HAV vaccination acceptance. The peaks in vaccination coverage rates in the Khabarovsk region in 2014 and 2020 are associated with floods in this region, followed by mass HAV vaccination in affected areas to prevent outbreaks. The annual HAV vaccination coverage rates in the Sverdlovsk region, Tuva, and Yakutia are significantly higher than the national average, suggesting adequate implementation of the vaccination programs in these regions. Interestingly, a trend toward a gradual decrease in annual coverage rates was observed in all three of these regions.

The ratio of adults and children among those vaccinated was uneven in different regions and in different years (Figure 5). In Tuva, almost 100% of those vaccinated were children under 18 years of age. In all other regions, the average proportion of children among those vaccinated varied from 29.3% in the Moscow region to 68.7% in the Sverdlovsk Region (Figure 5A). Analysis of each separate year did not reveal a trend toward an increase in the proportion of children among those vaccinated in Russia on average or in the Moscow and Khabarovsk regions (Figure 5B). Conversely, in Tuva, Yakutia, and the Sverdlovsk region, the proportion of children among those vaccinated annually remained high.

## 4. Discussion

HAV seroprevalence studies are an indispensable tool for assessing the proportion of the population susceptible to the virus and implementing prevention strategies that are largely based on the data these provide. In the current study, we analyzed the changes in HAV seroprevalence over the last decade following the start of a substantial decrease in reported incidence rates in regions of Russia where different HAV prevention strategies have been adopted. Moreover, these regions represent different federal districts of the Russian Federations, belong to climatic zones, and differ significantly in size, population density and urban/rural population ratio as shown in Figure 1. In two of the study regions, Moscow and Khabarovsk, no UMV program is in place. The same applies to Russia in general, where vaccinations are mainly provided to at-risk groups. Although the vaccination of children aged 3 to 6 years before entering kindergarten has been carried out in Moscow since 2009, our data on vaccination coverage in the region indicate that implementation of this vaccination policy has been minimal. Thus, the two above regions can be effectively labeled as regions “with no HAV child vaccination program.” Data on low vaccination coverage indicate that it is HAV circulation, rather than vaccination, that principally determines the level of population immunity in the Moscow and Khabarovsk regions. This remains the case for children and adolescents, where a significant proportion of seropositive participants may have acquired anti-HAV antibodies from past infection, considering the low vaccination coverage in these age groups. Thus, our data indicate the continued, albeit reduced, circulation of the virus in younger generations.

The decline in HAV seroprevalence in adolescents and younger adults, as reflected by the drop in reported incidence rates, and the shift in the age at the midpoint of population immunity to older age groups in the Moscow and Khabarovsk regions over the last decade are characteristic for regions undergoing a transition from high or intermediate to low endemicity worldwide [18,19,20]. These data are especially illustrative in the Moscow region, where archival data for the 1980s and 1990s are available. The endemicity of HAV is classified according to age-specific prevalence as high (≥90% by age 10 years), intermediate (≥50% by age 15 years with <90% by age 10 years), low (≥50% by age 30 years with <50% by age 15 years), and very low (≤50% by age 30 years) [21]. According to this system of classification, the Moscow region has transitioned from intermediate HAV endemicity in the 1980s, to low in the 1990s, to very low within the last two decades. In the absence of substantial HAV vaccination coverage, this decrease in HAV circulation results in a significant increase in the proportion of susceptible adolescents and adults, which brings with it the risk of hepatitis A outbreaks. Both foodborne HAV outbreaks among the general population and outbreaks of various scales in high-risk groups have recently been reported in regions where endemicity is low or shifting to low, and there is no UMV strategy or where vaccination coverage in adolescents and young adults is moderate [22,23]. The best way to prevent such outbreaks in the future seems to be the introduction of UMV to immunize children and adolescents before they reach the age of risk-associated behavior. The data from Spain, where different HAV vaccination strategies are in place in different regions of the country, clearly show that UMV has a significant impact on the cumulative incidence rate of outbreaks and the proportion of non-imported outbreaks compared to a vaccination strategy targeting risk groups [24].

There are only three regions in Russia where UMV programs are in place. In two regions (Yakutia and Sverdlovsk), the standard two-dose-based UMV scheme has been implemented since 2011 and 2003, respectively, and in Tuva single-dose child HAV immunization was launched in 2012. All of these regional programs resulted in significant changes in HAV seroprevalence in children and adolescents. In Yakutia, the existing vaccination program resulted in almost all age groups under 20 years reaching 50% herd immunity within the decade, compared to seroprevalence rates of 20–30% in 2008. In the Sverdlovsk region, where a UMV program has been in place for almost 20 years, HAV immunity has increased drastically over the last decade in children over 10 years, adolescents, and young adults compared to older age groups, in which immunity to HAV has declined. However, a three-fold drop in HAV immunity occurred in children under 10 years over the course of the decade, indicating a significant decline in vaccination coverage. In Tuva, eight years of child vaccination resulted in a two-fold increase in the level of HAV immunity in children under 10 years of age, which exceeded 50% by 2020. However, diminished HAV circulation in the region as reflected in incidence rates reported at zero in recent years has resulted in a significant decrease in population immunity among adolescents and adults under 40 years old in the region. Although highly efficient among children, the HAV vaccination campaign in Tuva caused a transition to intermediate-to-low endemicity in the region. Expansion of the vaccination program to adolescents is therefore required in order to maintain protection against the disease.

Vaccination coverage is one of the key factors determining the effectiveness of immunization programs. Unfortunately, data on exact rates of HAV vaccination coverage in the study regions, i.e., the percentage of those vaccinated among those who should be vaccinated, are limited. Thus, in the Sverdlovsk region, HAV vaccination coverage data are available only for 2009, when 65–70% coverage rates were reached in children aged 6–8 years [8]. In Tuva, coverage reached 87.4% in the first years following the launch of UMV [11]. However, data on vaccination coverage in subsequent years are not available. To overcome this data gap, we calculated coverage as the number of those vaccinated per 100,000 of the total or child population per year based on data for the total number of those vaccinated in each region. In regions that implemented HAV vaccination programs (Tuva, Yakutia, and Sverdlovsk), the calculated coverage rates in children do not exceed 8500–13,700 per 100,000 per year, which is about three-fold lower than hepatitis B virus (HBV) child vaccination coverage rates in these regions. These rates are 23,000 to 50,000 per 100,000 of the child population depending on the region, which constitutes a more than 95% reported HBV vaccination coverage rate [25]. Moreover, we observed a gradual decline in initially high HAV vaccination coverage rates in recent years in regions where child vaccination programs are in place. However, these data are not indicative of whether there is a true decline in vaccination coverage, as they reflect the number of those vaccinated among the total population of the region, but not among those who needed to be vaccinated. Thus, these declining calculated coverage rates may be associated with a true decrease in vaccination coverage, or they may reflect a gradual decrease in the proportion of those who should be vaccinated. Thus, the lack of coverage data reflecting the proportion of those vaccinated among those who should be vaccinated is a major limitation of our study. Nonetheless, HAV vaccination coverage rates in these three regions are the highest compared to the national average. Moreover, these vaccination rates, on average, are a thousand times higher than the HAV incidence rates in these regions, suggesting that it is vaccination, not infection, that determines seroprevalence in younger age groups. However, the decline in anti-HAV detection rates observed in the Sverdlovsk region in children under 10 years in 2021 compared to the 2008 data indicates a true decrease in vaccination coverage in that age group. There are two possible reasons for the decrease in HAV vaccination coverage rates. Firstly, vaccine supply issues and restrictions on the region’s budget for purchasing a vaccine may play a role. This issue was reported in relation to an HAV vaccination program in Panama, where the complete vaccination coverage rate in 2010 was reported to be only 40% due to vaccine supply issues [26]. Another reason for low coverage rates may be low levels of HAV vaccine acceptance due to vaccine hesitancy within the population and the belief, even among healthcare providers; that hepatitis A is not a serious illness; or the perception that the incidence of hepatitis A is declining in the community or that general precautions are sufficient to prevent transmission of the disease [27]. Moreover, the COVID-19 pandemic could result in a further decrease in HAV vaccination coverage in 2020–2021, as decline in routine vaccinations was noted globally during the first phase of the pandemic due to interrupted vaccination demand and supply, including reduced availability of the health workforce [28].

Apart from vaccination coverage, the proportion of the population with immunity is the main indicator of the immunological effectiveness of a vaccination program. In risk groups, the herd immunity (i.e., anti-HAV prevalence) threshold required to prevent HAV outbreaks was shown in the USA to be about 69% among homeless individuals or those who use drugs, with a critical vaccine threshold of 76% calculated from assumed 90% vaccine efficacy [29]. A similar estimate for the herd immunity threshold (65%) was obtained from a population of men who have sex with men in Australia [30]. Data from our study indicate, based on the example of Tuva, that HAV seroprevalence above 51% in children and adolescents may be enough to achieve protection against outbreaks in the general population. In this regard, the low anti-HAV detection rates (below 25%) observed in children under 10 years in the Sverdlovsk region in 2021 jeopardize the effectiveness of UMV in the region. The major reason for failure in implementation of regional HAV vaccination programs in Russia seems to be that these programs are financed from the regional budgets, but not from the federal budget. As a result, due to limited regional budgets and underestimation by regional governments of the medical significance of hepatitis A and the associated economic burden, sufficient amounts of the vaccine are not purchased. This problem can be solved by introducing hepatitis A vaccination into the National Immunization Schedule and purchasing the vaccine at the expense of the federal budget. Clearly, this will require an economic justification, and data from our study on the current epidemiology of hepatitis A can be used as the basis for these calculations.

## 5. Conclusions

We observed waning immunity to HAV in adolescents and young adults following the decrease in reported HAV incidence in regions where child vaccination programs are not in place. However, ongoing HAV circulation in these regions documented by looking at disease incidence and detectable antibodies in children and adolescents against a background of poor vaccination coverage suggests that UMV should be implemented on a national level. Existing regional HAV child vaccination programs, both double- and single-dose-based, demonstrated a significant impact on herd immunity to HAV. Our data on changes in HAV seroprevalence indicate that these regions may benefit from the expansion of HAV vaccination programs to include adolescents and young adults who were not vaccinated in childhood. Moreover, measures to control vaccination coverage and catch-up vaccination campaigns are recommended in order to maintain the effectiveness of existing HAV vaccination programs.

## Figures and Tables

**Figure 1 vaccines-10-01423-f001:**
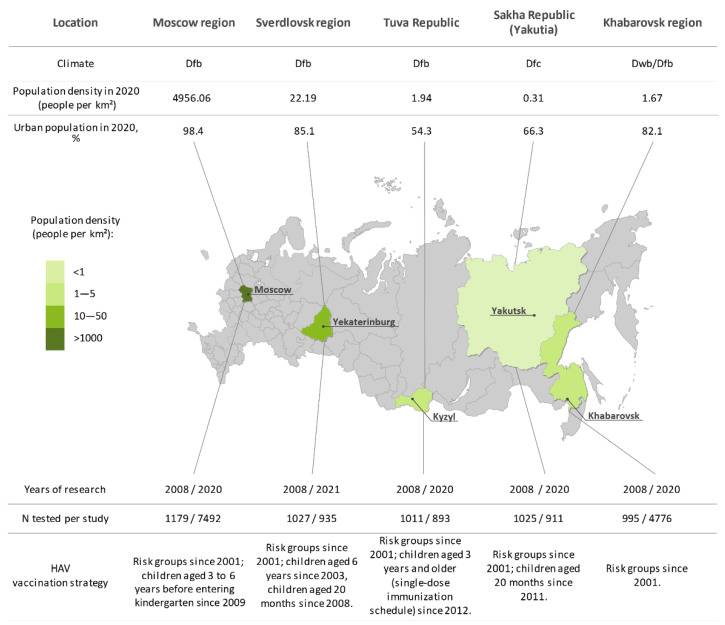
Study regions (in green) shown on a map of Russia alongside numbers of sampled sera in each year of the study, climate, sociogeographic conditions, and HAV vaccination strategy in each region. The color bar represents differences in population density between study regions. The capital of each study region is underlined. Social statistics for each region are taken from official statistical data. Climate zones are indicated in accordance with the Koppen classification (Dfb, warm summer continental, even annual rainfall; Dfc, continental subarctic, even annual rainfall; Dwb, warm summer continental, low winter rainfall).

**Figure 2 vaccines-10-01423-f002:**
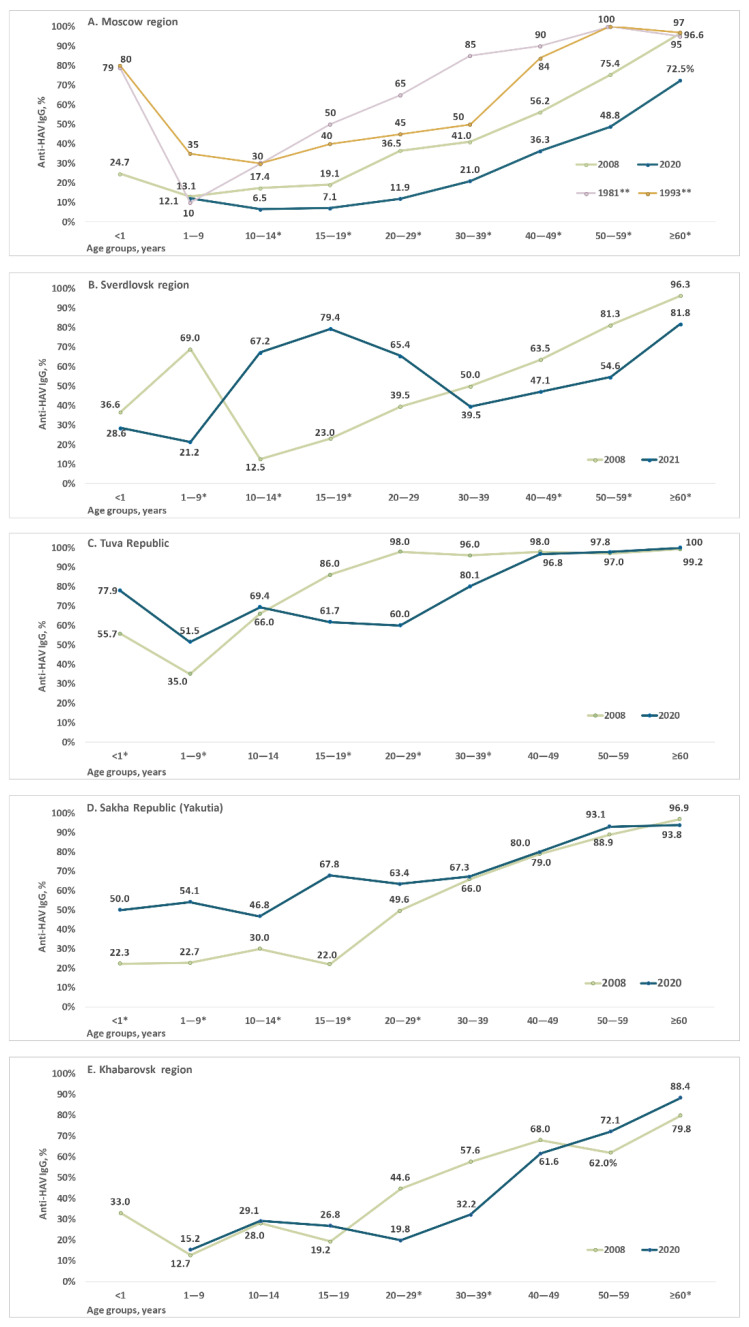
Proportions of serum samples positive for anti-HAV IgG antibodies in each age group of healthy individuals in the Moscow region (**A**), Sverdlovsk region (**B**), Tuva Republic (**C**), Sakha Republic (Yakutia) (**D**), and Khabarovsk region (**E**) Note: * *p* <0.05 (Fisher’s exact test) when data from 2008 and 2020/2021 are compared. ** Archived data on age-specific anti-HAV prevalence rates in the Moscow region in 1981 and 1993.

**Figure 3 vaccines-10-01423-f003:**
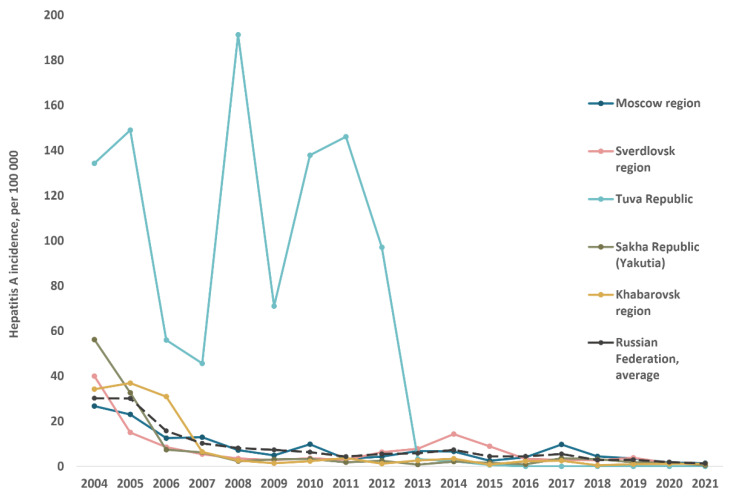
Hepatitis A reported annual incidence rates in total population of studied regions compared to the average in Russia, 2004–2021.

**Figure 4 vaccines-10-01423-f004:**
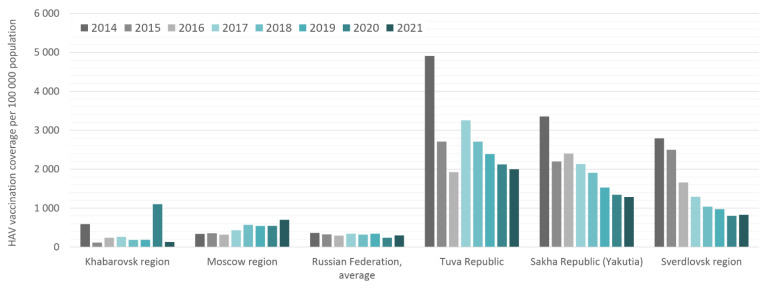
HAV vaccination coverage rates in the study regions compared to the Russian national average in 2014–2021. Vaccination coverage data are expressed as the number of those vaccinated per 100,000 per year.

**Figure 5 vaccines-10-01423-f005:**
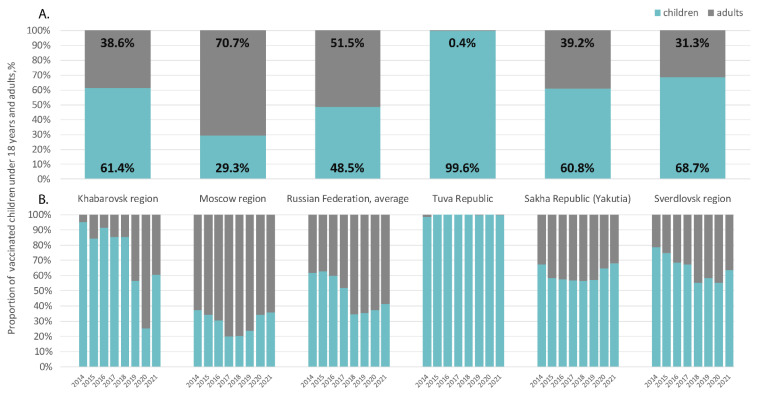
Ratios of children under 18 and adults among those vaccinated against HAV in the study regions compared to the Russian national average in 2014–2021, on average (**A**) and year by year (**B**).

**Table 1 vaccines-10-01423-t001:** Sociodemographic characteristics of participants screened for anti-HAV antibodies in 2008 and 2020/2021.

Study Region	Year of Study	Average Number of Samples per Age Group (Min.–Max.)	Male/Female Ratio	Urban/Rural Population Ratio
Moscow region	2008	131 (97–237)	1:1.1	1:0.09
	2020	936 (184–2751)	1:2.1	1:0.03
Sverdlovsk region	2008	114 (81–200)	1:2.4	1:0.02
	2021	103 (35–204)	1:1.6	1:0.01
Tuva Republic	2008	112 (88–200)	1:1.8	1:1.66
	2020	99 (64–169)	1:0.9	1:0.17
Sakha Republic (Yakutia)	2008	114 (97–203)	1:1.4	1:0.28
	2020	101 (58–185)	1:1.4	1:0.34
Khabarovsk region	2008	110 (99–197)	1:1.1	1:0.14
	2020	596 (294–1255)	1:0.9	1:0.05

**Table 2 vaccines-10-01423-t002:** Average anti-HAV IgG prevalence in total population, by gender and by place of residence.

Study Region	Year of Study	Anti-HAV IgG Antibody Prevalence							
		Average, % [95% CI]	*p* *	Men, % [95% CI]	Women, % [95% CI]	*p* *	Urban Population, % [95% CI]	Rural Population, % [95% CI]	*p* *
Moscow region	2008	40.1 [37.4–42.9]	**>0.0001**	38.1 [34.1–42.2]	41.9 [38.1–45.8]	0.1905	39.4 [36.5 –42. 5]	48.0 [38.3–57.8]	0.1068
	2020	23.0 [22.0–24.0]		22.0 [20.4–23.7]	23.5 [22.3–24.6]	0.1679	22.6 [21.6–23.6]	35.6 [21.6–23.6]	**>0.0001**
Sverdlovsk region	2008	54.0 [51.0–57.1]	0.856	43.6 [38.1–49.2]	58.4 [54.8–62.0]	**>0.0001**	54.0 [50.9–57.0]	66.7 [20.2–94.4]	1
	2021	54.6 [51.3–57.7]		53.3 [60.9–45.8]	54.8 [60.9–48.7]	0.6854	54.4 [51.2–57.6]	63.6 [35.2–85.0]	0.7625
Tuva Republic	2008	77.4 [74.7–79.8]	0.4162	70.1 [65.2–74.5]	81.5 [78.3–84.3]	**>0.0001**	70.3 [65.5–74.6]	81.6 [78.4–84.5]	**>0.0001**
	2020	75.7 [72.8–78.4]		67.4 [63.0–71.5]	85.0 [81.3–88.1]	**>0.0001**	76.9 [73.8–79.8]	68.7 [60.3–76.0]	**0.0474**
Sakha Republic (Yakutia)	2008	49.7 [46.6–52.7]	**>0.0001**	38.4 [33.9–43.1]	57.6 [53.6–61.4]	**>0.0001**	52.6 [49.1–56.0]	39.3 [33.1–45.8]	**0.0005**
	2020	67.1 [64.0–70.0]		71.2 [66.6–75.4]	63.8 [59.5–67.8]	**0.0193**	67.2 [63.5–70.6]	66.8 [60.5–72.6]	0.9356
Khabarovsk region	2008	41.8 [38.8–44.9]	0.0002	37.0 [32.7–41.4]	46.1 [41.9–50.4]	**0.0038**	43.1 [39.8–46.4]	32.5 [24.8–41.3]	**0.0299**
	2020	35.5 [34.1–36.9]		35.6 [33.7–37.4]	35.4 [33.4–37.4]	0.9275	38.1 [36.5–39.6]	39.3 [31.8–46.8]	0.8048

* Fisher’s exact test, significant differences (*p* < 0.05) are shown in bold.

## Data Availability

The data presented in this study are available in this article and its Appendix A.

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
