# Peer review of "Dynamic Changes in Hepatitis A Immunity in Regions with Different Vaccination Strategies and Different Vaccination Coverage"

_vaccines, 2022, doi:10.3390/vaccines10091423_

Round 1
Reviewer 1 Report
This comprehensive study examined the dynamic changes in hepatitis A immunity in several regions in Russia over the past decade. Based oh their analyses, the authors suggested the need for implementing a universal mass vaccination strategy on a national level. The analysis was well done and thorough, and the data are convincing. Thus, this is an important study that should be of great interest to epidemiologists and policy makers.
Author Response
Comment
This comprehensive study examined the dynamic changes in hepatitis A immunity in several regions in Russia over the past decade. Based on their analyses, the authors suggested the need for implementing a universal mass vaccination strategy on a national level. The analysis was well done and thorough, and the data are convincing. Thus, this is an important study that should be of great interest to epidemiologists and policy makers.
Response
We are very grateful to Reviewer for comment and appreciation of our study.
Reviewer 2 Report
This a well written manuscript, with clear and concise data. However some things are not clear to the reviewer:
"Thus, in Sverdlovsk Region, HAV vaccination coverage was reported 335 to have reached 65–70% in children aged 6–8 years by 2009 [7]. In Tuva, coverage reached 336 87.4% in the first years following the launch of UMV [10]. However, data on vaccination 337 coverage in subsequent years are not available. To overcome this data gap, we calculated 338 coverage as the number of those vaccinated per 100,000 of the total or child population 339 per year based on data for the total number of those vaccinated in each region. In regions 340 that implemented HAV vaccination programs (Tuva, Yakutia, and Sverdlovsk), the calcu- 341 lated coverage rates in children do not exceed 8,500–13,700 per 100,000 per year, which is 342 about three-fold lower than hepatitis B virus (HBV) child vaccination coverage rates in 343 these regions. "
what do the authors mean in the sentence above? that the official data are fake? Can the calculation be a bit more clear?
Also the article is mostly descriptive - what are the intended future health politics? What are the reasons behind the failure in implement the vaccination program and why it was somehow ok implemented in some regions and not others?
In figure 2 particularly Moscow data maybe some different colors to enhance contrast should be used.
Author Response
We are very grateful to Reviewer for comments and thorough analysis of our paper.
Comment 1
"Thus, in Sverdlovsk Region, HAV vaccination coverage was reported to have reached 65–70% in children aged 6–8 years by 2009 [7]. In Tuva, coverage reached 87.4% in the first years following the launch of UMV [10]. However, data on vaccination coverage in subsequent years are not available. To overcome this data gap, we calculated coverage as the number of those vaccinated per 100,000 of the total or child population per year based on data for the total number of those vaccinated in each region. In regions that implemented HAV vaccination programs (Tuva, Yakutia, and Sverdlovsk), the calculated coverage rates in children do not exceed 8,500–13,700 per 100,000 per year, which is about three-fold lower than hepatitis B virus (HBV) child vaccination coverage rates in these regions. "
what do the authors mean in the sentence above? that the official data are fake? Can the calculation be a bit more clear?
Response 1
With this sentence, we intended to stress that the official data on HAV vaccination coverage (shown as % of those who should be vaccinated) are very limited and are available for a single year per region only, usually soon after the start of vaccination campaign. To avoid the misunderstanding, we rewrote this sentence (page 11, lines 345-347 and line 356 in revised manuscript). Then we compared our calculated HAV vaccination coverage rates with official data on HBV vaccination coverage rates, as the latter is available and reported annually. This was done to stress that actual HAV vaccination coverage is much lower compared to HBV vaccination rates.
Comment 2
Also the article is mostly descriptive - what are the intended future health politics? What are the reasons behind the failure in implement the vaccination program and why it was somehow ok implemented in some regions and not others?
Response 2
The major reason for failure in implementation of regional HAV vaccination programs seems to be that the regional vaccination programs are financed from the regional budgets, but not from the federal budget. As a result, due to limited regional budgets, underestimation of the medical significance of hepatitis A and the associated economic burden, the sufficient number of doses of the vaccine are not purchased. This problem can be solved by introducing hepatitis A vaccination into the National Immunization Schedule and purchasing the vaccine at the expense of the federal budget. Obviously, this will require an economic justification, and data from our study on the current epidemiology of hepatitis A can be used as the basis for these calculations. We added these consideration to Discussion section of the revised manuscript (page 12, lines 396-404).
Comment 3
In figure 2 particularly Moscow data maybe some different colors to enhance contrast should be used.
Response 3
We changed colors in Figure 2A to highlight changes in HAV seroprevalence in Moscow Region over decades.
Reviewer 3 Report
This paper discusses different vaccination coverages and strategies brings changes in HAV infections in Russia areas. HAV is a virus that has been circulating in human society and causing issues in public health. The topic of the study is very significant and people in research areas and local health areas will be benefited from the data presented from the work. However, there are still some points for improvements:
1. Figure 1, you can use different colors and add a color bar to represent a specific factor that plays the most important role in this paper. For instance, from light green to dark green represents smaller to larger population, or a smaller to larger amount of vaccination coverage. This will give readers a more direct sense on the differences in these areas.
2. Please also discuss why you are not choosing other areas in Russia or describe more on how the five areas in the study represents different areas in Russia.
3. Please also mention the transmission of HAV in the introduction section, since something like from human to human, or mouse issue, are closely related to the factor of urban or rural area.
4. Since Moscow may have more international connections since it is a metropolitan city and the location is closer to many European countries with more floating/transient populations like travelers, for instance, will that affect something? For instance, if half travelers are from Europe, what’s their vaccination coverage? Stuff like that would be an interesting point to be discussed in your discussion section.
5. Also, are there any specific reasons that you chose to do the sampling in 2008 and 2020/2021?
6. Since 2020 and 2021 are the years that the world is facing the COVID-19 pandemic, will the pandemic bring any effects on your results? For instances, you may want to discuss if there’s any diversities in the COVID prevention strategies and how people’s life is affected differently.
Author Response
We are very grateful to Reviewer for comments and thorough analysis of our paper.
Comment 1
Figure 1, you can use different colors and add a color bar to represent a specific factor that plays the most important role in this paper. For instance, from light green to dark green represents smaller to larger population, or a smaller to larger amount of vaccination coverage. This will give readers a more direct sense on the differences in these areas.
Response 1
We added a color bar to Figure 1 to represent differences in population density in study regions. Data on vaccination coverage are not included in this Figure, as they are given in Results section in Fig.4.
Comment 2
Please also discuss why you are not choosing other areas in Russia or describe more on how the five areas in the study represents different areas in Russia.
Response 2
There are only three regions in Russia where UMV programs are in place – Yakutia, Tuva and Sverdlovsk regions. That was the reason to choose these regions for the study. Moscow and Khabarovsk regions do not have implemented UMV programs and were chosen as they represent regions that are the most distant from each other and the most different in socio-demographic characteristics. Besides, all these five regions are different in terms of climate, population density and proportion of urban/rural population and represent different federal districts of the Russian Federation. We added these considerations to Discussion section in revised manuscript (page 10, lines 284 –286, and line 322).
Comment 3
Please also mention the transmission of HAV in the introduction section, since something like from human to human, or mouse issue, are closely related to the factor of urban or rural area.
Response 3
We added the information on HAV transmission to Introduction section in the revised manuscript (page 2, lines 51 –56).
Comment 4
Since Moscow may have more international connections since it is a metropolitan city and the location is closer to many European countries with more floating/transient populations like travelers, for instance, will that affect something? For instance, if half travelers are from Europe, what’s their vaccination coverage? Stuff like that would be an interesting point to be discussed in your discussion section.
Response 4
Indeed, the proportion of travelers and migrants is high in metropolitan cities and can affect any data from seroprevalence studies. To avoid this potential sample bias, one of criteria to inclusion ito the study was the permanent residence in the study region (Materials and methods section, page 3, lines 119-120).
Comment 5
Also, are there any specific reasons that you chose to do the sampling in 2008 and 2020/2021?
Response 5
There were no specific reason to choose 2008 for the start of the study except for the drop of reported HAV incidence below 10 per 100,000 since 2008 on average across Russia and in all study regions except Tuva. This preceded by gradual decrease in HAV incidence rates in previous years. Such decrease in reported incidence was a clear indication to start HAV seroprevalence study that could reflect the change in disease epidemiology. The next study was intended to be conducted not earlier that a decade after the first study. Thus, the start of a substantial decrease in reported HAV incidence rates was a starting point for this study. We added this point to Discussion in revised manuscript (page 10, lines 282-283).
Comment 6
Since 2020 and 2021 are the years that the world is facing the COVID-19 pandemic, will the pandemic bring any effects on your results? For instances, you may want to discuss if there’s any diversities in the COVID prevention strategies and how people’s life is affected differently.
Response 6
Indeed, the COVID-19 pandemic could result in decrease in HAV vaccination coverage in 2020-2021, as decline in routine vaccinations was noted globally during the first phase of pandemic. We added this consideration to Discussion section (page 11, lines 380-384 of revised manuscript).